# Immune-Checkpoint Inhibitors in Platinum-Resistant Ovarian Cancer

**DOI:** 10.3390/cancers13071663

**Published:** 2021-04-01

**Authors:** Alice Indini, Olga Nigro, Csongor György Lengyel, Michele Ghidini, Angelica Petrillo, Salvatore Lopez, Francesco Raspagliesi, Dario Trapani, Shelize Khakoo, Giorgio Bogani

**Affiliations:** 1Medical Oncology Unit, Fondazione IRCCS Ca’ Granda Ospedale Maggiore Policlinico, Via Francesco Sforza 35, 20122 Milan, Italy; alice.indini@policlinico.mi.it; 2Medical Oncology, ASST Sette Laghi, Ospedale di Circolo e Fondazione Macchi, 21100 Varese, Italy; olga.nigro@istitutotumori.mi.it; 3Head and Neck Surgery, National Institute of Oncology Hungary, Ráth György u. 7-9, 1122 Budapest, Hungary; lengyel.csongor@gmail.com; 4Medical Oncology Unit, Fondazione IRCCS Ca’ Granda Ospedale Maggiore Policlinico, Via Della Commenda 19, 20122 Milan, Italy; Michele.ghidini@policlinico.mi.it; 5Medical Oncology Unit, Ospedale del Mare, Via E. Russo, and University of Study of Campania “L.Vanvitelli”, Via Pansini n.5, 80100 Naples, Italy; angelic.petrillo@gmail.com; 6Fondazione IRCCS Istituto Nazionale dei Tumori di Milano, Via Giacomo Venezian 1, 20133 Milano, Italy; salvatore.lopez@istitutotumori.mi.it (S.L.); raspagliesi@istitutotumori.mi.it (F.R.); 7European Institute of Oncology, IRCCS, New Drug Development for Innovative Therapies, Via Ripamonti 435, 20141 Milan, Italy; Dario.trapani@ieo.it; 8Gynecologic Oncology Unit, Department of Medicine, The Royal Marsden NHS Foundation Trust, Downs Road, Sutton SM2 5PT, UK

**Keywords:** platinum-resistance, ovarian cancer, immunotherapy, immune checkpoint inhibitors, new drugs

## Abstract

**Simple Summary:**

Patients with platinum-resistant ovarian cancer experience poor prognosis. No mature evidence supports the routine adoption of immunotherapy alone in this setting. However, the combination of immunotherapy with target therapies seems to be a promising option in patients with ovarian cancer. Ongoing trials are testing the combination between immune therapy and other target therapies, including PARP inhibitors, TKI, and anti-angiogenetic therapies. Further evidence is needed to assess the real impact and cost-effectiveness of immmunotherapic agents in platinum-resistant ovarian cancer.

**Abstract:**

Platinum-resistant ovarian cancer (OC) has limited treatment options and is associated with a poor prognosis. There appears to be an overlap between molecular mechanisms responsible for platinum resistance and immunogenicity in OC. Immunotherapy with single agent checkpoint inhibitors has been evaluated in a few clinical trials with disappointing results. This has prompted exploration of immunotherapy combination strategies with chemotherapy, anti-angiogenics, poly (ADP-ribose) polymerase (PARP) inhibitors and other targeted agents. The role of immunotherapy in the treatment of platinum-resistant OC remains undefined. The aim of this review is to describe the immunobiology of OC and likely benefit from immunotherapy, discuss clinical trial data and biomarkers that warrant further exploration, as well as provide an overview of future drug development strategies.

## 1. Introduction

Ovarian cancer (OC) is the eighth most common cancer in women. Worldwide the OC incidence and mortality figures rank just below those for cancers of the cervix and uterus, accounting for 295,414 new cases and 184,799 deaths in 2018 [1,2]. In the United States (US), 21,750 new cases and 13,940 OC-related deaths are projected to occur in 2020. OC is amongst the leading causes of cancer-related death in women, particularly in the 40–79 year age group. Survival rates vary significantly depending on the disease stage at diagnosis. Ninety percent of patients with early-stage disease are alive at 10 years compared with only around 15% with advanced disease, despite optimal treatment [3]. Over 50% of patients present with advanced disease due to a lack of effective screening measures and the absence of specific symptoms which leads to diagnostic delay [4]. Most OC arises from the epithelial tissue (90%), and most of them seems originating from the fallopian tube [5]. The following histologic subtypes of epithelial OC are recognized according to the World Health Organization (WHO) classification: serous (70%), endometrioid (10%), clear cell (10%), and other types including mucinous tumors (10%). This classification was updated in 2014, with a distinction being made between two variants of serous carcinoma (i.e., high-grade serous carcinoma (HGSC), and low grade), characterized by distinct oncogenic pathways, differences in therapeutic approach, and prognosis. According to the latest WHO classification, high-grade cancers are all included in the HGSC group, due to similar biologic characteristics and prognosis [5]. HGSC is characterized by genomic instability and p53 mutations, while low-grade tumors tend to be more indolent and show dysregulation in the mitogen-activated-kinases (MAPK), with mutations predominantly observed in KRAS and BRAF and a few in other RAS proteins and MEK [5].

Although the therapeutic landscape and hence survival outcomes have improved for OC over the years with the inclusion of agents such as poly (ADP-ribose) polymerase 1 (PARP) inhibitors and the anti-angiogenic bevacizumab, surgery and platinum-based chemotherapy remains the mainstay of treatment. While initial response rates to chemotherapy can be high, most patients relapse and have or eventually develop resistance to platinum-based chemotherapy which confers a poor prognosis with a median overall survival (OS) of <12 months. There is therefore a clear unmet need to improve outcomes in this subset of patients.

Immune-checkpoint inhibitors (ICI) have revolutionized the treatment paradigm of several tumors such as melanoma and cutaneous squamous cell carcinoma, non-small cell lung cancer (NSCLC), urinary tract carcinomas, and head and neck cancers, and their use in other tumor types remains under investigation. The most widely adopted immunotherapy strategies in clinical practice include targeting the cytotoxic T-lymphocyte antigen 4 (CTLA-4), and the programmed cell death protein 1 (PD-1) and its ligand (PD-L1). However, not all tumors display the same degree of sensitivity to immunotherapy [6]. In OC patients, immunotherapy might be a promising option as adjuvant therapy (as in the VITAL trial) and for the treatment of recurrent/progressive disease [7]. This review aims to analyze the current landscape of immunotherapy in OC, focusing mainly on the setting of platinum-resistant OC. We provide an overview of the immunogenicity of OC and explore potential strategies to overcome immune-resistance in this disease.

## 2. Platinum-Resistant OC Models

### 2.1. Definitions of Platinum-Sensitivity, and Resistance

The platinum-free interval (PFI) is associated with prognosis and has been defined by the Gynecologic Cancer InterGroup (GCIG) consensus statement as the time from the last dose of platinum until documented disease progression. In patients with recurrent OC, the disease is defined as refractory, platinum-resistant, partially platinum-sensitive, or platinum-sensitive by a PFI of ≤4 weeks, <6 months, 6–12 months, or >12 months respectively [8]. This classification has been conventionally accepted also by the ESMO–ESGO consensus [9] and is used in most clinical trials as well as in clinical decision-making for treatments [10,11]. It should be mentioned that the definition of “platinum partially sensitive” OC has not been fully validated and utilized variably and inconsistently across clinical trials. Notably, the resistant phenotype can arise at different stages of the natural history of OC. Tumor cells may present with de novo or intrinsic resistance to platinum drugs referred to as primary platinum-resistance (PPR) or become resistant during therapy, known as acquired platinum-resistance (APR) [7,8,9,10,11,12,13,14,15,16,17].

### 2.2. Molecular Mechanisms of Platinum Resistance: A Continuum of Platinum-Resistance and Immunogenic Phenotypes?

Mechanisms of platinum-resistance are multi-factorial and include genetic and epigenetic alterations, apoptotic escape, and environmental factors [17]. Interestingly, associations between molecular mechanisms responsible for platinum resistance and immunogenicity in OC have been suggested, though mostly derived from speculative retrospective analyses and exploratory statistical exercises. Accordingly, a possible overlap of selected mechanisms conferring platinum-resistance, and immunogenicity might be just inexistent, resulted from misinterpretations of uncontrolled datasets, and should not be used to justify any bio-plausibility in this regard that platinum-resistant OC is tout court immune-responsive.

Cognizant of all the speculative nature of the assumptions, some mechanisms associated with platinum-resistance have been reported to confer some features of immune-response [17,18]. In this regard, key mechanisms possibly associated with sensitivity to immune-agents have been correlated to a platinum-resistant phenotype of OC. The loss of the tumor-suppressor gene AT-rich interaction domain A (ARID1A), a multidrug resistance-associated protein 2 (MRP2) regulator, has been shown to lead to a platinum-resistant phenotype in a preclinical tumor model [19,20,21]. The loss of function of ARID1A has also been associated with molecular alterations in DNA repair mechanisms and increasing mutation frequency, similar to those observed with the loss of mismatch repair (MMR). ARID1A-deficient ovarian cells present with a microsatellite instability (MSI) genomic signature with an increased mutational burden, along with elevated numbers of tumor-infiltrating lymphocytes, and PD-L1 expression, thereby providing a rationale to explore the role of immunotherapy in this setting. Indeed, in a murine model of OC, ARID1A-deficient tumors showed a higher response rate and improved survival when exposed to anti-PD1 ICIs [20]. The loss of MMR proteins (i.e., MMR deficiency), which are responsible for correcting single-strand DNA errors, leads to MSI, either by genetic inheritance or somatically acquired. MLH1 acquired deficiency related to gene promoter methylation, or MLH2 mutation, are the most frequent causes of MSI in OC [21,22,23,24,25,26,27,28,29]. Besides, the survival mechanism to overcome replication arrest due to DNA damage is performed by several DNA polymerases, including POLH. An elevated baseline expression of POLH in OC stem cells has been linked to intrinsic chemo-resistance [30,31,32,33]. Interestingly, both MMR deficiency and alterations in POL-proteins have been associated with platinum-resistance and an immune-responsive phenotype, therefore representing speculatively potential biomarkers for immunotherapy response in a subgroup of OC.

Germline and somatic mutations in the BRCA1 and 2 genes have been identified in 17% and 6% of patients with HGSC respectively, and are associated with improved response to both platinum compounds and other DNA targeting agents such as PARP inhibitors [34]. The presence of BRCA mutations has also been associated with a mutagenic phenotype with increased neoantigens [35,36,37,38,39,40]. One study used The Cancer Genome Atlas dataset and reported a potential role of BRCA alterations to predict immunogenicity in HGSC, with higher immunotherapy-responsive signatures in patients harboring the BRCA1-mutation; however, only newly diagnosed tumors were considered [34].

A discussion of all concurrent mechanisms contributing to platinum-resistance in OC has been extensively reviewed elsewhere and is beyond the scope of this review [17]. However, an improved understanding of the molecular mechanisms underlying resistance has propelled the development of therapeutic targets both in this setting and as a means to delay the onset of chemoresistance [38]. More recently, the recognition that platinum-resistant OC has some features indicative of an immune responsive phenotype has led to an interest in evaluating immunotherapy in this context [39,40]. However, all the evidence reported is speculative, and the immune-context of platinum-resistant OC seems to be heterogeneous, and not immediately resulting in an immunogenic phenotype. The reductio ad unum of the platinum-resistant phenotype to an immunogenic cancer type is currently inappropriate, unbiased, and largely speculative. However, a subgroup of treatment-resistant OC, reliably a tiny subgroup, may present molecular stigmata associated with immune-response.

## 3. The Current Management of Platinum-Resistant OC: Few Signs of Progress in Therapies, Few Options for Clinical Care

The current treatment paradigm for platinum-resistant OC involves the sequential use of single-agent chemotherapy and avoids compounds with cross-resistance to platinum. Single-agent chemotherapy provides similar responses to combination treatment with less toxicity. Paclitaxel, gemcitabine, topotecan, and liposomal anthracyclines have shown response rates ranging from 10% to 15%, with progression-free survival (PFS) of ~3 months [41,42]. The attempts to implement targeted agents in this setting provided small or no added benefits in the treatment, including the antiangiogenic drugs bevacizumab and cediranib [43,44,45,46,47].

Continuous low-dose chemotherapy administration (i.e., metronomic schedules) has also been investigated in platinum-resistant OC, based on possible anti-angiogenic and speculated immune-modulating activities [47,48]. Findings across studies have consistently shown that the therapeutic armamentarium for platinum-resistant OC is associated with only modest benefit. There is therefore a clear need to improve outcomes in this subset of patients and this should be considered a research priority.

## 4. The Immune Contexture and Immunogenicity in OC

OC is characterized by a high copy number variation (CNV). These alterations included deletion of genes implicated in homologous recombination (BRCA), base excision repair, and DNA damage signaling. The role of immunotherapy in OC is still controversial. In particular, platinum-resistant OC should not be considered an “immunogenic disease”, but some subtypes may present features associated with better benefits from immunotherapy agents. In some patients with platinum-resistant disease, markers of an antitumor immune response can be detected in peripheral blood, tumor tissue, and ascites fluid [49,50]. Like in several different solid tumors, the presence of tumor-infiltrating lymphocytes (TILs) in the tumor microenvironment (TME) of OC is associated with improved survival, as showed in studies on tumor samples from patients with OC [51]. Conversely, the lack of TILs is associated with worse survival. However, distinct immune cell subtypes can be identified within the TILs, playing different roles. Intra-tumoral CD3 +TILs were associated with improved survival in advanced stage OC in a report by Zhang et al. [52]. A study by Sato et al. [53] showed that intraepithelial CD8+, rather than CD3+ TILs, were associated with a favorable prognosis. A meta-analysis including 1815 OC patients confirmed the prognostic value of intraepithelial CD8+ TILs, regardless of tumor grade, stage, or histological subtype although it should be noted that heterogeneity between studies was significant [54]. The presence of infiltrating B lymphocytes has been associated with improved prognosis [55]. Moreover, evidence suggests that the presence of both B cells and CD8+ TILs is associated with increased survival in OC patients, compared to CD8+ TILs alone [56].

However, in the tumor microenvironment (TME), multiple factors counteract the activity of TILs, thereby facilitating cancer progression through immune escape mechanisms. Regulatory T lymphocytes (Tregs) play a fundamental role in mechanisms of immune escape and evasion [57,58]. Tumor-associated macrophages (TAMs), one of the major subpopulations of myeloid cells in OC ascites, promote an immune-suppressive milieu through reduced IL-12 production, as shown in a preclinical in-vitro and ex vivo model [59,60,61]. TAMs promote mechanisms of immune suppression, favoring cancer growth, angiogenesis, and metastasis. Other subtypes of immune-suppressive cells include the natural killer (NK), and the myeloid-derived suppressor cells (MDSCs) which inhibit both adaptive and innate immunity through several complex mechanisms. Like TAMs and Tregs, MDSC negatively impacts patient survival by enhancing cancer progression and metastasis [62].

Together with immune-suppressive cells, several inhibitory molecules can dampen the CTL-mediated antitumor response within the OC TME. These immune pathways involve the programmed cell death-ligand 1 (PD-L1), the cytotoxic T lymphocyte antigen 4 (CTLA-4), the lymphocyte activation gene 3 (LAG-3) protein, and the T cell immunoglobulin and mucin domain-containing protein 3 (TIM-3) [63,64,65]. In platinum-resistant OC, TILs often express PD-1, while malignant cells can show high levels of PD-L1 [66,67]. A study comparing the prognosis of patients with OC according to PD-L1 expression showed that patients with high PD-L1 have significantly worse prognoses when compared with those showing lower PD-L1 expression [68]. An inverse correlation was observed between PD-L1 expression and the CD8+ TILs count, suggesting that PD-L1 expression on tumor cells can directly suppress antitumor CD8+ T cells. In multivariate analysis, the expression of PD-L1 on tumor cells and intraepithelial CD8+ T-lymphocyte counts were independent prognostic factors. However, there is a lack of concordance on this topic among different studies [69,70], mostly due to different staining methods, type of cells used, score for surface PD-L1 expression (tumor cell vs. immune infiltrate vs. both), and the optimal cut-off to define PD-L1 positivity.

OC cells can evade the host immune system through several other mechanisms, including loss of MHC expression [71] and up-regulation of immunosuppressive factors, such as TGF-β [72] indoleamine 2,3-dioxygenase (IDO) [73], and cyclooxygenases (COX-1 and COX-2) [74].

Although evidence suggests there might be a rationale for immunotherapy in platinum-resistant OC, the complex interaction between OC cells and cells of the immune system may partly explain the limited benefit observed with immune-agents in this disease, as well as the non-reproducibility of some findings. In general, platinum-resistant OC seems not a primary immune-responsive cancer type, although some of these tumors can present immunogenic features possibly deriving a benefit from pharmacological manipulations of the immune response or conferring better prognosis.

For example, many of the key original works showing some prognostic meaning of TILs were based on tissue samples obtained before any chemotherapy, therefore not capturing, or adjusting for key prognostic covariates. Moreover, these findings are largely unconfirmed in the clinical setting and derive from in vitro and animal models, and some retrospective patient series This is a major limitation and may explain the divergencies in the literature on this topic, with different conclusions. Therefore, data suggesting a prognostic role of immune-related biomarkers in platinum-resistant OC are to be considered generally inconclusive, essentially unconfirmed.

## 5. Immunotherapy for Platinum-Resistant or Refractory OC

### 5.1. Trials Investigating Single-Agent Immune-Checkpoint Inhibitors

To date, only a few clinical trials have investigated immune checkpoint inhibitors (ICI) as a treatment for advanced recurrent OC [75]. Table 1 displays the characteristics and results of the principal clinical trials of immunotherapy in OC.

A phase II study of the anti-CTLA4 antibody ipilimumab in patients with platinum-sensitive recurrent OC (*n* = 40), reported an ORR of 10.3%, with a high incidence of treatment-related adverse events (AEs) (i.e., 50% of patients with grade 3 AEs) [76]. The high rate of toxicity might be related to the dose of ipilimumab administered per protocol, which is much higher than the approved schedule of ipilimumab for other tumors (e.g., melanoma). After this initial experience with ipilimumab, there have been few trials investigating anti-CTLA4 in recurrent OC. Most trials have focused on anti-PD1/PD-L1 therapy.

Brahmer et al. reported the results of the phase I trial of the anti-PD-L1 antibody BMS-936559 (MDX-1105) in solid tumors (regardless of PD-L1 expression) and included 17 patients with recurrent OC [77]. The disease control rate (DCR) in this small cohort was 23%, and the median duration of response (DOR) was 1.3 months. The first phase II trial exploring two different schedules of nivolumab (1 mg/kg or 3 mg/kg every 2 weeks) in 20 patients with platinum-resistant OC showed more promising results. The ORR was 15%, with a DCR of 45% [77]. The median PFS was 3.5 months (95% CI, 1.7 to 3.9 months), and the median OS was 20 months (95% CI, 7 months to not reached). Eighty percent of patients had high PD-L1 expression in their tumor tissue, however, PD-L1 expression was not found to be associated with ORR [78]. In the KEYNOTE-028 phase I trial which included PD-L1 positive OC (*n* = 26), treatment with single-agent pembrolizumab resulted in an ORR of 11.5%; 7 patients (26.9%) achieved SD as the best response [78]. Median PFS and OS were 1.9 (95% CI, 1.8–3.5) and 13.8 (95% CI, 6.7–18.8) months, respectively [78]. The subsequent phase 2 KEYNOTE-100 trial explored single-agent pembrolizumab in two different cohorts of recurrent OC patients: cohort A enrolled 285 patients who had received ≤3 prior lines of treatment, with a treatment-free interval (TFI) of 3–12 months; cohort B enrolled 91 patients, who had received 4–6 prior lines with a TFI of ≥3 months [79]. ORR was 7.4% for cohort A and 9.9% for cohort B, with a median DOR of 8.2 months for cohort A and not reached for cohort B; DCR was 37.2% and 37.4%, in cohort A and B, respectively. Response rates differed according to PD-L1 expression, using the combined positive score (CPS), with an ORR of 4.1% for CPS < 1, 5.7% CPS ≥ 1, and 10.0% for CPS ≥ 10. PFS was 2.1 months for both cohorts (95% CI 2.1–2.2 for cohort A and 95% CI 2.1–2.6 for cohort B); the median OS was not reached for cohort A (95% CI 16.8-not reached) and was 17.6 months for cohort B (95% CI 13.3-not reached). The protocol-specified final analysis of this trial has recently been presented at the American Society of Clinical Oncology (ASCO) annual meeting, confirming a trend toward better ORR and longer OS in both study cohorts with higher PD-L1 expression [80].

Recently, the JAVELIN trial evaluated another anti-PD-L1 antibody (avelumab) in recurrent OC. The JAVELIN trial was a phase 1b trial enrolling patients with several solid tumors, with an expansion cohort that assessed efficacy outcomes in patients with recurrent OC unselected for PD-L1 expression (*n* = 124) [81]. Preliminary results from this trial showed an ORR of 9.7%, and a DCR of 54% in patients with OC. Patients with PD-L1 positive tumors (≥1% tumor cell staining), accounting for 77% of all patients, had better ORR compared with patients with PD-L1 negative tumors (12.3% vs. 5.9%, respectively). Overall, median PFS was 11.3 weeks (95% CI: 6.1–12.0) and median OS was 10.8 months (95% CI: 7–16.1). Data regarding safety of single-agent anti-PD1/PD-L1 in recurrent OC mirror those of melanoma and NSCLC trials, confirming the overall good tolerability of treatment and the low rate of severe (i.e., grade 3 according to CTCAE v 4.0) treatment-related adverse events AEs (~10–15%) [81,82,83,84,85,86,87,88,89,90,91,92]. The most common immune-related AEs (irAEs) of any grade occurring in 10% of patients across trials were: fatigue (which is also the most common overall toxicity), diarrhea, nausea, increased lipase, skin reactions, and thyroid dysfunction. Overall, the observed rate of treatment discontinuation due to treatment-related AEs was low [81,82,83,84,85,86,87,88,89,90,91,92].

Despite a clear rationale for investigating immunotherapy in OC, results from the aforementioned clinical trials of single-agent ICIs in recurrent and/or resistant OC have been disappointing. Efforts to improve treatment efficacy have focused on combining immunotherapy with cytotoxic chemotherapy or targeted agents including anti-angiogenic or combining different ICIs.

### 5.2. Trials Investigating Combination Treatment with ICIs and a Discussion of the Rationale

One of the most promising combination strategies is co-targeting PARP and PD-1 [82]. There is a strong rationale in support of this strategy, given that homologous recombination deficient (HRD) tumors show high expression of PD-1, and the preclinical evidence that double-strand DNA break inducing drugs such as PARP inhibitors allow the accumulation of mutations and hence neoantigens, stimulate upregulation of PD-L1 in tumor cells and activate the innate immune system via the STING pathway with type-I interferon production resulting in optimal recruitment of dendritic cells and priming of T effector cells [83].

Results from two combination trials of PARP inhibitors and anti-PD1/PD-L1 antibodies have recently been presented [84,85]. The TOPACIO/KEYNOTE-162 [85] was a single-arm phase 1/2 study of niraparib in combination with pembrolizumab, in women with advanced/metastatic triple-negative breast cancer (TNBC) or recurrent OC, irrespective of BRCA mutation status. Analysis of the pooled OC cohort (*n* = 62 patients) showed an ORR of 18% (90% CI, 11–29%), with a DCR of 65% (90% CI, 54–75%). Subgroup analysis revealed that ORRs were consistent regardless of platinum-sensitivity, previous bevacizumab treatment, or tumor BRCA or HRD biomarker status. The MEDIOLA trial [84] evaluated the combination of olaparib and durvalumab in patients with platinum-sensitive ROC with known BRCA mutation (*n* = 34). Preliminary results of this trial showed an ORR of 72%, and a 12-week DCR of 81%. Response rates were higher in patients who had received only one prior line of chemotherapy and were not associated with PD-L1 expression. The safety of combination treatment was assessed in both trials: the most common treatment-related AEs of any grade were fatigue, nausea, anemia, and constipation. The incidence of irAEs in the TOPACIO trial was 19%, with grade 3 irAEs occurring in 6% and no reports of treatment-related deaths. More recently, the results of combination treatment with the anti-CTLA4 antibody tremelimumab and the PARP inhibitor olaparib were presented at ASCO 2020 [86] Fifty percent of patients receiving combination treatment were platinum-resistant and 75% had no BRCA mutation. The combination treatment was associated with poor activity with 8% achieving a partial response (*n* = 1), 25% having stable disease (*n* = 3), and 17% having a PFS of >6 months (*n* = 2).

Another notable strategy has been combination therapy with anti-angiogenic and PD-1/PD-L1 inhibitors [87]. The rationale behind this approach is to dampen the immunosuppressive tumor microenvironment to enhance the immune response. VEGF blockade increases immune cell infiltrate through the promotion of T-cell trafficking thereby decreasing the ratio of MDSCs and Tregs. The combination of the anti-PD-L1 antibody atezolizumab with bevacizumab was shown to reduce the progression of platinum-resistant OC in vivo through the suppression of epithelial-mesenchymal transition [87].

A phase 2 study investigated combination treatment with nivolumab and bevacizumab and recruited 38 patients of which 18 (47%) had platinum-resistant disease [88]. The ORR in the trial population was 28.9% (95% CI, 15.4–45.9%) with a much higher ORR of 40% (95% CI, 19.1–64.0%) in platinum-sensitive patients compared with 16.7% (95% CI, 3.6–41.4%) in platinum-resistant patients. Interestingly, the proportion of patients with PD-L1 positive tumors was higher amongst platinum-resistant patients and so this cannot account for the difference in activity observed. It is difficult to interpret the data with immune-agents plus anti-vascular drugs (and in general with combination therapies) from uncontrolled clinical and/or single-arm trials, what agent provided a benefit, and if the combination is truly synergistic. Anti-vascular agents alone, in fact, can provide some benefit in patients with OC, as previously discussed. Such a question can find an answer only in controlled randomized clinical trials, and should not prompt over-enthusiastic conclusions, as well as discourage the design of clinical trials of the combination regimens without arms enrolling to single agents, as control.

Dual blockade with the checkpoint inhibitors ipilimumab and nivolumab [93] has demonstrated increased anti-tumor activity compared to either single agent alone in other tumor types where it is a standard of care treatment option. Both of these agents exert their effects through distinct pathways during T cell activation and therefore a combination strategy is justified. The recently published phase II randomized NRG oncology study recruited patients with a PFI of <12 months and included 100 patients. Most patients had platinum-resistant disease: 63.3% in the nivolumab monotherapy arm and 60.8% in the ipilimumab/nivolumab combination arm. The ORR within 6 months was significantly higher in the combination arm (31.4% vs. 12.2%) as was the median PFS (3.9 vs. 2 months, HR, 0.528; 95% CI, 0.339 to 0.821; *p* = 0.004). However, combination treatment did not result in a statistically significant increase in OS, and toxicity was higher as expected but comparable to previous reports (grade 3 treatment-related AEs: 49% vs. 33%). An exploratory analysis revealed that poor prognostic markers including platinum resistance, worse performance status, older age, more previous lines of therapy, obesity, and higher baseline tumor burden favored the combination arm.

The combination of chemotherapy and immunotherapy is another strategy under investigation. Chemotherapy induces an immune-mediated mechanism of tumor-killing (immunogenic death). Anthracyclines, taxanes, and platinum compounds can activate antineoplastic immune responses, enhancing the recognition of the altered self-material [94]. However, the use of the hypomethylating agent (guadecitabine), to prime and enhance tumor cell recognition by CD8+ cells, before treatment with pembrolizumab only resulted in poor anti-tumor activity in patients with platinum-resistant OC [95]. Interestingly, preliminary data suggest that neoadjuvant platinum-based chemotherapy has an impact on the immune cell composition of OC [89]. Following platinum-based chemotherapy, there was a significant increase in CD4+ (*p* = 0.03) and CD8+ infiltration (*p* = 0.009) and a decrease in FOXP3+ cells (*p* = 0.01). This provides further supporting evidence for combining chemotherapy with ICIs. However, the results of the randomized, phase III, JAVELIN 200 trial [89] were presented at the Society of Gynecologic Oncology 2019 meeting and included 566 patients with platinum-resistant ovarian cancer (25% platinum-refractory), were disappointing. Patients were randomized to avelumab, liposomal doxorubicin, or a combination of the two. The combination arm was not associated with any significant survival benefit. Interestingly, there was a higher ORR in the combination arm amongst the PDL1 positive subgroup (18.5%, 95% CI 11.1–27.9) compared to the PDL1 negative subgroup (3.4%, 95% CI 0.4–11.9). There was also a trend towards an improvement in PFS (3.7 vs. 3.0 months, HR 0.65, 95% CI 0.46–0.92) and OS (17.7 vs. 13.1 months, HR 0.72, 95% CI 0.48–1.08) when the combination treatment was compared to liposomal doxorubicin alone in PDL1 positive patients which accounted for 57%. We need to better understand which patients are most likely to benefit from particular treatment strategies.

### 5.3. Identification of Biomarkers of Immune-Response

Along with efforts to improve therapeutic outcomes by evaluating combination strategies, there is a strong need for biomarkers to guide patient selection. Evidence suggests that response to immunotherapy might be higher in certain subtypes of OC. For example, ~10% of clear cell tumors are MSI-H, and consequently, show higher PD-1 expression rates compared to their serous counterparts [90]. Similarly, mutations in BRCA1/2 and TP53, which confer a significant lifetime risk for ovarian carcinoma, correlate with a higher neoantigen load [91]. In patients with OC, both BRCA1/2 and TP53 mutation status correlate with increased PD-1/PD-L1 expression levels, and they might therefore be more likely to respond to immunotherapy [92]. High PD-1/PD-L1 expression on tumor cells and TILs has been associated with a favorable prognosis in HGSC [96].

Several biomarkers of response to immunotherapy, which have been studied in other solid tumors, have also been investigated in OC and are described in detail below:PD-L1: PD-L1 expression, assessed through immune-histochemistry techniques, has been investigated as a biomarker to predict response to anti-PD1 therapy in several tumors. In some cases, the indication for immunotherapy either as a single agent or in combination with other agents depends on the degree of PD-L1 expression on tumor tissue (e.g., NSCLC). In HGSC, PD-L1 expression has been reported in 90% of cases, with 30% deemed to have a high expression of the biomarker [91,92]. However, to date the data regarding the role of PD-L1 as a marker to predict response to immunotherapy in HGSC are inconsistent. Moreover, different methods to define PDL1 status have been used across studies which makes interpreting the overall value of this potential biomarker more challenging [91].TILs: As discussed, the presence of abundant TILs in tumor tissue is associated with favorable clinical outcomes in several solid tumors including HGSC [97,98,99]. TILs modulate the tracking and response to neoantigens and play a role in reducing resistance to platinum compounds. However, the presence of abundant TILs per se is not sufficient to predict response to immunotherapy, since different mechanisms acting in tumor cells and within the TME can affect the action of TILs and reduce the immune response [98]. Moreover, recent evidence from multi-region analysis of metastatic sites suggests that even a single metastatic site with relative immune privilege may lead to treatment resistance despite immune response elsewhere [99].TMB: TMB is defined as the total number of somatic coding mutations in a tumor. Highly mutated tumors are more likely to produce tumor-specific epitopes, acting as neoantigens that are recognized as non-self by the immune system [100]. Tumors with an increased TMB are potentially more immunogenic and may therefore benefit from immunotherapy. A correlation between high TMB and improved clinical response in HGSC has been reported as well [101]. A retrospective analysis revealed that the presence of BRCA mutations and high TMB was associated with longer OS in patients with HGSC [101]. However, prospective data confirming the role of TMB as a potential biomarker are still awaited. Results of the phase II trial KEYNOTE-158 (NCT02628067), which investigated the use of pembrolizumab in patients with solid tumors and high TMB, provide preliminary data in this setting. The Food and Drug Administration (FDA) has recently approved the supplemental Biologics License Application for pembrolizumab as treatment of adult and pediatric patients with unresectable or metastatic solid tumors with high TMB (i.e., >10 mutations/Mb) [99].

In the context of platinum-resistant HGSC, the most compelling evidence is the interplay between genetic instability and immune response, which supports the combination of drugs targeting DNA repair processes and immunotherapy as the most promising future combination strategy. However, none of the biomarkers here presented have been prospectively validated in controlled clinical trials and none is prime time for clinical use. Therefore, validation in the context of randomized controlled clinical trials is warranted.

### 5.4. Clear Cell Ovarian Carcinoma

Clear-cell ovarian carcinoma (CCOC) represents a distinct entity of OC, associated with a unique clinical and genetic pattern [102] When diagnosed in the early stage, CCOC is generally associated with a better prognosis than HGSC; however, in the advanced setting, the prognosis of CCOC is dismal, with poor response to standard treatments [102] The genomic landscape of CCOC is characterized by recurrent pathogenic dysregulation of the PIK3CA-AKT-PTEN-mTOR pathway in nearly half of all cases. CCOC may occur in the context of Lynch syndrome, most commonly as a result of the germline mutations MSH2 and MLH1 [103]. Besides, half of CCOC present deleterious ARID1A mutations, which is a suggested mechanism of resistance to multiple anticancer agents associated with response to various immunotherapy agents. In experimental models, ARID1A mutated CCOC has a higher TMB with increased TILs and enhanced susceptibility to anti-PDL-1 [104] However, no study molecularly-selecting for this subtype of patients has been reported. Subgroup analysis from the KEYNOTE-100 clinical trial showed an improved response in patients with CCOC (*n* = 19 patients), with an ORR of 15.8% (CI 95% 3.4–39.6), with one complete response [79]. Though not statistically significant for this small population, the survival analysis showed a trend towards an improved OS for CCOC, suggesting that immunotherapy should be further investigated in this setting. The NRG Oncology [93] study also provided further support for immunotherapy as patients with CCOC (12% of the study population) had fivefold odds of response compared to other histologic subtypes. Taken together, this tiny corpus of evidence should be carefully viewed and considered no more than anectodical reports. It is nebulous whether the clear cell histology is the determinant of the immunogenicity or more likely such a benefit is driven by the MSI status, that is enriched in this subtype of OC.

To date, several clinical trials of immunotherapy for patients with CCOC are ongoing, but none have been reported. Interestingly, one study is investigating the role of nivolumab with or without ipilimumab for the treatment of advanced extra-renal clear cell carcinomas (NCT03355976), based on the concept that the tumorigenesis of clear cell variants may confer immunogenic properties [105]. Both renal and ovarian clear cell carcinomas present common alterations of the DNA remodeling complex SWI–SNF, as well as PI3K-mTOR dysregulations, suggesting that drug development might be tumor agnostic for the clear cell histotype.

Pending the results of ongoing clinical trials, a significant proportion of CCOC patients will be eligible for immunotherapy with pembrolizumab, based on the approved indication of MMR deficient neoplasms and/or tumors with high TMB-of which CCCOs seem to be enriched [106].

## 6. Future Directions for Immune Checkpoint Inhibitor Combinations in Platinum-Resistant or Refractory OC

Most ongoing clinical trials include anti-PD-1/PD-L1 antibodies as backbone ICIs [107,108,109] (Table 2). Due to a strong rationale, combination therapy with immunotherapy and PARP inhibitors continues to be of interest with three clinical trials underway: the MOONSTONE trial (NCT03955471), the BOLD trial (NCT04015739), and the OPAL trial (NCT03574779). Interestingly, the combination of immunotherapy with tyrosine kinase inhibitor (TKI) seems associated with promising results (as reported in the phase II LEAP trial). Similar to the rationale behind immunotherapy and PARP inhibitor combinations, radiotherapy can also induce DNA damage and modulate the TME. The combination of immunotherapy with radiotherapy as a possible treatment strategy has been extensively reviewed elsewhere.

Results from studies combining two drugs have not been as successful as anticipated. Therefore the triple combination strategy of chemotherapy, immunotherapy, and targeted agents (either PARP or VEGF inhibitors) is currently under investigation in several ongoing trials [109]. Immunotherapy with agents targeting the mitogen-activated protein kinases (MAPKs) pathway, which is dysregulated in 3–11% of OC patients, is another strategy that has gained interest [110,111]. The immunomodulatory impact of MAPK inhibition has been demonstrated in a wide range of tumors [111]. Several studies have shown that MEK inhibitors increase the expression of intrinsic and IFN-γ-induced HLA/MHC I and II in cancer cells, and the number of CD8+ TILs [111]. The combination of MEK inhibitors with ICIs is under investigation. The BEACON trial (NCT03363867) is currently ongoing to assess the efficacy and safety of an anti-PD-L1 monoclonal antibody combined with a MEK-inhibitor and an anti-VEGF monoclonal antibody. Notably, the majority of the ongoing trials are designed and sponsored by companies, and few explore therapies based on early clinical or preclinical evidence matured in independent research contexts. This can introduce some elements of bias in the design, data reporting, and interpretation of the results, including in the selection of the regimens in the control arms (when is not a placebo) or the declination of the medical unmet needs into uncontrolled studies—the ones that leave ample margins of uncertainties and more often expose patients to therapies with doubtful value. This has wide potential consequences, from suboptimal clinical care driven by hyped results to broader societal harms.

Given the immunosuppressive tumor micro-environment in platinum-resistant OC and the poor benefit observed with single immunotherapy agents, most current research focuses on enhancing immunotherapy response with the use of combination treatments. Although beyond the scope of this review, it is important to note that other immunotherapy strategies under investigation include adoptive cell therapy and vaccines. Evidence from ongoing clinical trials will hopefully change the treatment paradigm of platinum-resistant disease in the next future.

## 7. Conclusions

Evidence from pre-clinical and clinical studies suggests that at least a subgroup of OC patients show a pro-active immune contexture, favoring a response to immunotherapy, either alone or in combination with other agents. However, to date, there are no established biomarkers to select OC patients that are likely to respond to immunotherapy. Available data on the prognostic and predictive significance of PD-1 and PD-L1 expression, as well as the presence of TILs, in OC, are inconsistent. Prospective controlled trials are currently underway to identify potential biomarkers of response or resistance to immunotherapy. Treatment options remain limited for patients with platinum-resistant OC. Therefore, clinical trials that aim to identify key drivers of immune-response and optimize patient selection to improve future outcomes should be considered a priority. Presently, the only place for immune-checkpoint inhibitors in platinum-resistant OC is through well-designed clinical trials.

## Figures and Tables

**Table 1 cancers-13-01663-t001:** Key clinical trials of immunotherapy for recurrent ovarian carcinoma.

Trial Name, Identification Number	Study Design	Drug(s)	Disease Setting	Sample Size	ORR	mPFS, mo (95% CI)	mOS, mo (95% CI)	TRAEs G ≥3 Incidence
NCT01611558	Phase II	Ipilimumab 10 mg/kg q3w per 4 cycles; maintenance q12w	Platinum-sensitive	*n* = 40	10.3%	-	-	50%
NCT00729664	Phase I	BMS-936559 3 mg/kg–10 mg/kg q2w	Recurrent disease	*n* = 17	6%	-	-	ND *
UMIN000005714	Phase II	Nivolumab 1 mg/kg–3 mg/kg q2w	Platinum-resistant	*n* = 20	15%	3.5 (1.7–3.9)	20.0 (7.0−NR)	40%
KEYNOTE-028, NCT02054806	Phase Ib	Pembrolizumab 10 mg/kg q3w	PD-L1 + recurrent disease	*n* = 26	11.5%	1.9 (1.8–3.5)	13.8 (6.7–18.8)	3.8%
KEYNOTE-100, NCT02674061	Phase II	Pembrolizumab 200 mg q3w	Recurrent disease (cohort A: PFI 3−12 mo; cohort B: PFI ≥3 mo)	*n* = 376	8%	A: 2.1 (2.1–2.2) B: 2.1 (2.1–2.6)	A: NR (16.8–NR) B: 17.6 (13.3–NR)	19.7%
JAVELIN, NCT01772004	Phase Ib	Avelumab 10 mg/kg q3w	Recurrent disease	n =125	9.6%	10.2 (5.4–16.7) **	11.2 (8.7–15.4)	7.2%
TOPACIO, NCT02657889	Phase I/II	Niraparib 200 mg QD + Pembrolizumab 200 mg q3w	Recurrent disease	*n* = 62	18%	3.4 (2.1–5.1)	-	6%
MEDIOLA, NCT02734004	Phase I/II	Olaparib 400 mg BID + Durvalumab 1500 mg q3w	Platinum-sensitive gBRCAm	*n* = 34	72%	-	-	-
NCT02485990	Phase I/II	Tremelimumab 10mg/kg q4wx7 then q12w alone or +Olaparib 150mg BID or Tremelimumab 3 mg/kg q4wx7 then q12w + Olaparib 150 mg BID	Recurrent disease	*N* = 24	8% ^†^	17% PFS > 6mo	-	42%
NCT02873962	Phase II	Bevacizumab 10 mg/Kg + Nivolumab 240 mg q2w	Recurrent disease	*N* = 38	28.9%	9.4 (6.7–NA)	-	23.7%
NCT02498600	Phase II	Nivolumab 3 mg/kg q2wx4 or Nivolumab 3 mg/Kg + Ipilimumab 1 mg/Kg q3wx4, maintenance nivolumab 3 mg/kg q2wx42 max	Recurrent disease	*N* = 100	Nivo: 12.2% Ipi/Niv:31.4%	Nivo:2 Ipi/Nivo:3.9 (HR 0.53; 0.34–0.82)	Nivo:21.8 Ipi/Nivo:28.1 (HR 0.79; 0.44–1.42)	Nivo:33% Ipi/Nivo:49%

(source: www.clinicaltrials.gov, accessed on 1 February 2020). * incidence of adverse events was assessed in the whole trial population and not specifically in the ovarian cancer subpopulation of patients (for whole data on safety, refer to Brahmer, J.R.; Tykodi, S.S.; Chow, L.Q.; et al. Safety and activity of anti-PD-L1 antibody in patients with advanced cancer. *N. Eng. J. Med.*
**2012**; *36618*: 2455–2465, ** 1-year progression free survival, ^†^ ORR amongst patients receiving combination treatment, BID, bis in die; gBRCAm, germline breast related cancer antigen mutant; mo, months; mOS, median overall survival; mPFS, median progression free survival; ND, not determined; NR, not reached; ORR, overall response rate; PFI, platinum-free interval; QD, once daily; q2-3-12w, every 2–3–12 weeks; TRAEs, treatment-related adverse events.

**Table 2 cancers-13-01663-t002:** Ongoing clinical trials of immune-check point inhibitors in recurrent platinum-resistant ovarian cancer.

Experimental Regimens	ICI Target	Biomarkers for Patient Selection	Exploratory Biomarkers of Benefit	Phase	NCT Identifier
Camrelizumab, apatinib.	PD-1	-	-	2	NCT04068974
Dostarlimab (TSR-042), niraparib	PD-1	-	-	2	NCT03955471 (MOONSTONE)
* multiple arms of durvalumab combinations	PD-L1	HDR, other *	-	2	NCT03699449 (AMBITION)
Pembrolizumab, chemotherapy	PD-1	-	PD-L1	2	NCT03539328 (MITO27)
Pembrolizumab, Lenvatinib	PD-1	-	PD-L1	2	NCT03797326 (LEAP-005)
Durvalumab, TPIV200/huFR-1 #	PD-L1	-	-	2	NCT02764333
Atezolizumab, Bevacizumab, ASA	PD-L1	-	-	2	NCT02659384
Atezolizumab, Bevacizumab, cobimetinib	PD-L1	-	-	2	NCT03363867 (BEACON)
Durvalumab, AVB-S6-500	PD-L1	-	^ phenotypic change in immune-cells	1/2	NCT04019288
Utomilumab **	4-1BB (CD137)	PRAME, COL6A3 (CAA) HLA-A *0201 (HLA)	-	1	NCT03318900
Durvalumab, Bevacizumab, Olaparib	PD-L1	-	TMB, HR, TII	2	NCT04015739 (BOLD)
Dostarlimab (TSR-042), niraparib, bevacizumab	PD-1	-	-	2	NCT03574779 (OPAL)
Durvalumab, ONCOS-102~	PD-L1	-	-	1/2	NCT02963831
Durvalumab, azacitidine	PD-L1	-	-	2	NCT02811497 (METADUR) $
Durvalumab, focal radiotherapty ^^	PD-L1	-	-	1	NCT03283943
Pembrolizumab, ENB003	PD-1	-	PD-L1, ETBR	1/2	NCT04205227
Emactuzumab, bevacizumab, Paclitaxel	CSF1R	-	Radiomic parameters ##	2	NCT02923739
Nivolumab, Pembrolizumab, DSP-7888	PD-1	HLA-A* 02:01, HLA-A* 02:06, HLA-A*24:02	-	1/2	NCT03311334
Avelumab, TRX518, Cyclophosphamide	PD-L1, GITR	-	-	1/2	NCT03861403

Immune-checkpoint inhibitors (ICI) are reported in italic. Data extracted from Clinicaltrial.gov (last access 23 March 2020). PD-1, programmed death 1. PD-L1, programmed death-ligand 1. HDR, Homologous Recombination Deficiency. ASA, acetyl salicylic acid. CCA, cancer-associated antigens. HLA, Human Leucocyte Antigen. TMB, Tumour Mutational Burden. HR, Homologous recombination status. TII, tumour immune infiltrate. ENB003, Endothelin B Receptor Antagonist. ETBR, Endothelin B Receptor. CSF1R, Colony stimulating factor 1 receptor. DSP-7888, WT1 protein-derived peptide vaccine. GITR, glucocorticoid-induced TNFR-related protein. PRAME, Melanoma antigen preferentially expressed in tumors. COL6A3, collagen type VI alpha 3 chain. * umbrella Study of Biomarker-driven Targeted Therapy with olaparib + cediranib or durvalumab + olaparib or durvalumab + chemotherapy or durvalumab + tremelimumab + chemotherapy or durvalumab + tremelimumab + paclitaxel treatment. The study enrolls patients to PAPR-inhibitors combination treatments if HDR tumors or other biomarker-based allocation (not specified) for non-HDR tumors. # Multi-Epitope Anti-Folate Receptor Vaccine ^changes in T cell populations (including but not limited to CD3, CD8, CD4, FOXP3) and cell proliferation and changes in the proportion of macrophage phenotypes M1 and M2 (with phenotypic markers potentially including arginase1, CD11b, PDL-1, and CD206) ** adoptive immunotherapy with transferred central memory-type CTL targeting ovarian cancer antigens administered alone, and in combination with, utomilumab ~ONCOS-102 is an oncolytic adenovirus armed with human GM-CSF and an Ad5/3 chimeric capsid. $ Basket Study, ^^ 24 Gray (6 Gy X 4 fractions), may be escalated to 32 Gy (8 Gy per 4 fractions). ## non-invasive imaging macrophage-specific imaging, ADC (apparent diffusion coefficient) for cellularity, and DCE (dynamic contrast enhanced) for vasculature.

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
