# Peer review of "Immune-Checkpoint Inhibitors in Platinum-Resistant Ovarian Cancer"

_cancers, 2021, doi:10.3390/cancers13071663_

Round 1

Reviewer 1 Report

This review paper examines immune checkpoint inhibitors in ovarian cancers with platinum resistance and particularly on the association and/or overlap with immunogenicity and immunotherapy. It covers published clinical trials and studies of the clinical, molecular, biological aspects in a well-ordered structure. It is a comprehensive and detailed review, particularly focusing on the variety of treatment combinations (checkpoint inhibitors, chemotherapy, PARP inhibitors etc). Different tumour types/histopathology/biomarkers are also reviewed. Clinical trials results / ongoing trials are tabulated and examined in depth. The authors clearly state when data is lacking or conclusions are not based on solid evidence, and point out when papers are inconsistent on a particular issue. This authors achieves their goal of reviewing the published literature on platinum resistant ovarian cancer and possible immunogenic and immunotherapy pathways.

The paper is written to a high standard though there may be some formatting adjustments listed below. I can find no major or even minor issues with this review. In summary, 1 or 2 minor adjustments may round out this excellent review article, though it is very acceptable in its present form.

On line 117: The sentence could be improved for readability, for example:
 The loss of function of ARID1A has also been associated with molecular alterations in DNA repair mechanisms and increasing
 mutation frequency, similar to those observed with the loss of mismatch repair (MMR)

Line 127 should rather mean something like "either by genetic inheritance or somatically acquired", not "both" as they are referring to mutually exclusive terms.

both genetic inheritance or somatically acquired.

In line 380:  Is there a missing reference for this section - it refers to "another study" ?: 

In another study, preliminary data suggest that neoadjuvant platinum-based chemotherapy has an impact on the immune cell composition of OC. Following platinum-based chemotherapy,there was a significant increase in CD4+ (p=0.03) and CD8+ infiltration (p=0.009) and a decrease in FOXP3+ cells (p=0.01). This provides further supporting evidence for combining chemotherapy with ICIs.

Possible formatting problem:

For example "grade ³ 3" instead of "grade 3"? Similarly on line 291: 

in ³ 10% of patients

Sometimes the reference citation is listed after the period.  See lines 140, 145

. (35-40)One study used The Cancer Genome Atlas

Author Response

The paper is written to a high standard though there may be some formatting adjustments listed below. I can find no major or even minor issues with this review. In summary, 1 or 2 minor adjustments may round out this excellent review article, though it is very acceptable in its present form.

Comment 1: On line 117: The sentence could be improved for readability, for example:
 The loss of function of ARID1A has also been associated with molecular alterations in DNA repair mechanisms and increasing
 mutation frequency, similar to those observed with the loss of mismatch repair (MMR)

Answer: in order to comply with the reviewer's comment, we corrected the sentence, accordingly. 

Comment 2: Line 127 should rather mean something like "either by genetic inheritance or somatically acquired", not "both" as they are referring to mutually exclusive terms.

Answer: in order to comply with the reviewer's comment, we corrected the sentence, accordingly. 

Comment 3: In line 380:  Is there a missing reference for this section - it refers to "another study" ?: 

In another study, preliminary data suggest that neoadjuvant platinum-based chemotherapy has an impact on the immune cell composition of OC. Following platinum-based chemotherapy,there was a significant increase in CD4+ (p=0.03) and CD8+ infiltration (p=0.009) and a decrease in FOXP3+ cells (p=0.01). This provides further supporting evidence for combining chemotherapy with ICIs.

Answer: in order to comply with the reviewer's comment, we corrected the sentence, accordingly. 

Comment 4: Possible formatting problem:

For example "grade ³ 3" instead of "grade 3"? Similarly on line 291: 

in ³ 10% of patients

Sometimes the reference citation is listed after the period.  See lines 140, 145

(35-40)One study used The Cancer Genome Atlas

Answer: in order to comply with the reviewer's comment, we corrected the sentences.  

Reviewer 2 Report

"Immune-checkpoint inhibitors in platinum-resistant ovarian cancer" by Indini et al provides a timely and comprehensive summary of the potential utility of immune-checkpoint inhibitors in the context of platinum resistant disease. The review is well-written and thoughtful, providing not just a summary of the field but a synthesis of many biological and clinical observations. The authors highlight key clinical and biological observations and highlight areas where more research is necessary (particularly with regards to the biological rationale for utilizing checkpoint inhibitors for platinum resistant disease). I have only a few minor points that I believe would help provide more context for future readers of the manuscript.

-Section 4: It could be helpful here to provide some context on the genomics of ovarian cancer, particular in the context of homologous recombination defects and antigenicity (as mentioned regarding combination therapies with PARP inhibitors in subsequent sections).

-To best contextualize the review for a general audience, a paragraph in the Section 5 that summarizes available immunotherapies and biomarkers/mechanisms related to response (in general) would provide helpful context for understanding the review

Author Response

Comment 1: Section 4: It could be helpful here to provide some context on the genomics of ovarian cancer, particularly in the context of homologous recombination defects and antigenicity (as mentioned regarding combination therapies with PARP inhibitors in subsequent sections).

Answer: In order to comply with the reviewer's comment, we added data on genomic landscape of OC at the beginning of section 4.

Comment 2: To best contextualize the review for a general audience, a paragraph in Section 5 that summarizes available immunotherapies and biomarkers/mechanisms related to response (in general) would provide helpful context for understanding the review

Answer: In order to comply with the reviewer's comment, we clarified and summarized in Tables the most innovative treatment modalities and biomarkers. 

Reviewer 3 Report

Overall this is a well written review that summarises an important topic, I have only minor comments : 

-line 53-authors should mention the fact the the epithelial origin is likely to be fallopian tube 

-line 86 the definition of platinum resistance has been updated in the most recent ESMO/ESGO consensus 

-it would be good to mention some of the ongoing adjuvant studies 

Author Response

Comment 1: line 53-authors should mention the fact the epithelial origin is likely to be fallopian tube 

Answer: In order to comply with the reviewer's comment, we corrected the sentence. 

Comment 2: line 86 the definition of platinum resistance has been updated in the most recent ESMO/ESGO consensus 

Answer: In order to comply with the reviewer's comment, we modified the text accordingly. 

Comment 3: it would be good to mention some of the ongoing adjuvant studies

Answer:  The aim of the present paper is to investigate the role of immunotherapy in patients with platinum-resistent OC. We did not focus our analysis on adjuvant therapies.However, in order to comply with the reviewer's comment, we mentioned the role of immunotherapy within the adjuvant setting.  

Reviewer 4 Report

Immune-checkpoint inhibitors in platinum-resistant ovarian cancer
Indini A, Nigro O, ….Bogani G. submitted to Cancers

Dr. Indini and colleagues provide a useful and generally up to date review of the application of immunotherapy in ovarian cancer with emphasis on immune checkpoint inhibitors in platinum resistant disease. The overall conclusions are not particularly exciting with respect to the impact of immunotherapy thus far; however, the authors hold out hope for combined therapy with ICIs and other targeted agents, radiotherapy or chemotherapy. 
This paper is well written and has evaluated the rather sparse set of trial results thus far fairly and without undue enthusiasm or lack thereof. The conclusions are balanced and appear to be appropriate.
Perhaps I missed it, but did the authors discuss the LEAP-005 interim trial results presented at the EMSO virtual conference, 2020 (abstract LBA41)? The combination of the multi-TKI lenvatinib + pembrolizumab had an ORR of 32% and a disease control rate of 74% at interim analysis in highly pretreated ovarian cancer patients. While the authors note the potential of improvement of ICIs when combined with bevacizumab, specifically inhibiting VEGFA, more attention should be paid to the other targets of multi-TKIs beyond VEGF and the mechanisms of additive or synergistic effects of these combinations. What are the broader impacts of TKIs on the TME and the tumor cells themselves with respect to PD-L1 expression? 
I suggest adding a section to this manuscript highlighting more mechanistic hypotheses that could be explored in the translational endpoints of current and future immunotherapy trials. This would be an important and unique contribution to the literature beyond what is currently published. What molecular endpoints do the authors recommend be studied given their extensive review of the literature with respect to mechanisms? Yes, we agree that currently, there are no validated biomarkers, but please suggest some potential biomarkers for study.

Author Response

Comment 1: Perhaps I missed it, but did the authors discuss the LEAP-005 interim trial results presented at the EMSO virtual conference, 2020 (abstract LBA41)? The combination of the multi-TKI lenvatinib + pembrolizumab had an ORR of 32% and a disease control rate of 74% at interim analysis in highly pretreated ovarian cancer patients. While the authors note the potential of improvement of ICIs when combined with bevacizumab, specifically inhibiting VEGFA, more attention should be paid to the other targets of multi-TKIs beyond VEGF and the mechanisms of additive or synergistic effects of these combinations. What are the broader impacts of TKIs on the TME and the tumor cells themselves with respect to PD-L1 expression? 

Answer: In order to comply with the reviewer's comment, we addressed this point 

Comment 2: I suggest adding a section to this manuscript highlighting more mechanistic hypotheses that could be explored in the translational endpoints of current and future immunotherapy trials. This would be an important and unique contribution to the literature beyond what is currently published. What molecular endpoints do the authors recommend be studied given their extensive review of the literature with respect to mechanisms? Yes, we agree that currently, there are no validated biomarkers, but please suggest some potential biomarkers for study.

Answer: In order to comply with the reviewer's comment, we highlighted the importance of biomarkers in this setting. 

Round 2

Reviewer 4 Report

Now acceptable for publication in my opinion.  Reviewer's comments adequately addressed.